# Beta-Hydroxybutyrate Augments Oxaliplatin-Induced Cytotoxicity by Altering Energy Metabolism in Colorectal Cancer Organoids

**DOI:** 10.3390/cancers15245724

**Published:** 2023-12-06

**Authors:** Tolga Sever, Ender Berat Ellidokuz, Yasemin Basbinar, Hulya Ellidokuz, Ömer H. Yilmaz, Gizem Calibasi-Kocal

**Affiliations:** 1Department of Translational Oncology, Institute of Health Sciences, Dokuz Eylul University, 35340 Izmir, Turkey; 2Department of Internal Diseases, Gastroenterology, Faculty of Medicine, Dokuz Eylul University, 35340 Izmir, Turkey; 3Department of Translational Oncology, Institute of Oncology, Dokuz Eylul University, 35340 Izmir, Turkey; 4Department of Preventive Oncology, Institute of Oncology, Dokuz Eylul University, 35340 Izmir, Turkey; 5Department of Biology, The David H. Koch Institute for Integrative Cancer Research at MIT, Massachusetts Institute of Technology, Cambridge, MA 02139, USA; ohyilmaz@mit.edu; 6Broad Institute of MIT and Harvard, Cambridge, MA 02142, USA; 7Department of Pathology, Massachusetts General Hospital and Harvard Medical School, Boston, MA 02114, USA

**Keywords:** colorectal cancer, organoid, beta-hydroxybutyrate, oxaliplatin, reactive oxygen species, metabolic targeted therapy

## Abstract

**Simple Summary:**

One of the hallmarks of cancer is the deregulation of cellular metabolism. The study investigates the potency of the administration of beta-hydroxybutyrate (BOHB) to elevate oxaliplatin’s cytotoxic effect in colorectal cancer. The study employed an in vitro organoid model to assess this dual treatment. The results exhibited that non-toxic doses (7.5 and 10mM) of BOHB with oxaliplatin treatment significantly increased the cytotoxic effect of oxaliplatin in colorectal cancer organoids by altering energy metabolism, leading to higher levels of reactive oxygen species (ROS). Opposingly, healthy colon organoids were not affected by the dual treatment as in colorectal cancer organoids. Melatonin was found to neutralize this effect by protecting cancer cells from oxidative stress. The study suggests that BOHB may improve the effectiveness of chemotherapy, particularly with drugs that have a similar mechanism of action to oxaliplatin, in treating colorectal cancer, potentially leading to better outcomes for patients.

**Abstract:**

Deregulation of cellular metabolism has recently emerged as a notable cancer characteristic. This reprogramming of key metabolic pathways supports tumor growth. Targeting cancer metabolism demonstrates the potential for managing colorectal cancer. Beta-hydroxybutyrate (BOHB) acts as an acetyl-CoA source for the tricarboxylic acid (TCA) cycle, possibly redirecting energy metabolic pathways towards the TCA cycle that could enhance sensitivity to oxaliplatin, through the generation of reactive oxygen species (ROS). This study explores the potential of BOHB to enhance oxaliplatin’s cytotoxic effect by altering the energy metabolism in colorectal cancer. The study employed advanced in vitro organoid technology, which successfully emulates in vivo physiology. The combination treatment efficacy of BOHB and oxaliplatin was evaluated via cell viability assay. The levels of key proteins involved in energy metabolism, apoptotic pathways, DNA damage markers, and histone acetylation were analyzed via Western Blot. ROS levels were evaluated via flow cytometer. Non-toxic doses of BOHB with oxaliplatin significantly amplified cytotoxicity in colorectal cancer organoids. Treatment with BOHB and/or melatonin resulted in significantly decreased lactate dehydrogenase A and increased mitochondrial carrier protein 2 levels, indicating inhibited aerobic glycolysis and an increased oxidative phosphorylation rate. This metabolic shift induced apoptotic cell death mediated by oxaliplatin, owing to high levels of ROS. Melatonin counteracted this effect by protecting cancer cells from high oxidative stress conditions. BOHB may enhance the efficacy of chemotherapeutics with a similar mechanism of action to oxaliplatin in colorectal cancer treatment. These innovative combinations could improve treatment outcomes for colorectal cancer patients.

## 1. Introduction

According to GLOBOCAN data, approximately 1.9 million new cases of colorectal cancer (CRC) were diagnosed in 2020, culminating in nearly 935,000 deaths. Remarkably, CRC is the third most common cancer by incidence and the second by mortality [1]. Carcinogenesis involves multiple mutations, with specific key mutations identified as significant drivers of this process. In the context of CRC initiation and progression, the inactivation of tumor suppressor genes, like P53 and adenomatous polyposis coli (APC), and the activation of oncogenes, such as the Kirsten rat sarcoma viral oncogene homolog (KRAS), is crucial [2]. CRC treatment integrates a multidisciplinary approach, amalgamating surgery, radiation, and chemotherapy. Oxaliplatin, a platinum-based chemotherapy drug, is commonly used in CRC management. As a third-generation diaminocyclohexane platinum compound, oxaliplatin forms intra-strand adducts, disrupting DNA transcription and replication, and ultimately triggering apoptotic cell death [3]. Neoadjuvant use of oxaliplatin post-surgery has demonstrated modest single-agent activity, with an efficacy of around 20–24% in phase trials. However, when combined with 5-FU and leucovorin, oxaliplatin has been shown to improve progression-free survival and overall survival, outperforming regimens that use 5-FU and leucovorin alone [4,5,6]. Despite its primary mechanism of action being to induce DNA damage, leading to apoptotic cell death, the full range of oxaliplatin effects remains to be fully elucidated. Furthermore, given that oxaliplatin is a non-selective chemotherapeutic agent, normal cells are also exposed to its cytotoxic effects. Notably, when used clinically, oxaliplatin can lead to a neurotoxic side effect known as peripheral neurotoxicity [7]. To prevent such adverse reactions in patients with colorectal cancer, it is imperative to consider dose reductions.

Numerous cellular alterations mark the transformation into cancer cells during carcinogenesis, with each change serving as a unique cancer hallmark [7]. One such hallmark is the modification of energy metabolism, often characterized by the Warburg effect, first identified by Otto Warburg. This effect indicates the preference of many cancer types for aerobic glycolysis, a glycolytic pathway utilized even in the presence of oxygen. In contrast, normal cells typically resort to oxidative phosphorylation for energy generation in mitochondria. However, cancer cells often reroute their energy metabolism towards glycolysis, resulting in increased glucose uptake and lactate production. This shift provides several advantages to cancer cells, including enhanced nutrient availability, drug resistance, and accelerated cell proliferation [8].

Metabolic changes in cancer cells often lead to higher levels of reactive oxygen species (ROS) than their normal counterparts [9]. While moderate ROS levels can induce DNA mutations and inflammation, promoting carcinogenesis, exceedingly high ROS levels can instigate cancer cell death. Notably, CRC cells typically present higher ROS levels than healthy cells, suggesting that ROS induction might offer a viable strategy for inducing apoptosis in CRC.

In CRC, mitochondrial activity is often suppressed, limiting ROS production by electron transport chain (ETC) components. However, mitochondria respond to ketones, which convert to acetyl-CoA and undergo catabolism within the mitochondrial matrix. Hence, beta-hydroxybutyrate (BOHB), the main circulating ketone body, was chosen to inhibit the glycolytic pathway and amplify the tricarboxylic acid (TCA) [10] cycle. To examine ROS-dependent cell death, we included melatonin, known for its protective role under high oxidative stress conditions, in the treatment groups as an additional adjuvant [11].

Organoid technology was employed due to its superior ability to recapitulate the complex cellular architecture and physiology of tissues in vivo. Unlike traditional two-dimensional culture systems, which can oversimplify the cellular environment, organoids retain key aspects of the original tissue’s morphology, functionality, and genetic features. This includes the intricate network of cell–cell and cell–matrix interactions, which is crucial for understanding tumor behavior and response to treatment. Therefore, the use of organoid technology can provide deeper insights and more clinically relevant data in cancer research [12].

In the present study, the primary objective was to modulate the energy metabolism of cancer cells by employing BOHB. By favoring the TCA cycle, it was hypothesized that BOHB administration could potentially augment the effectiveness of the chemotherapeutic agent, oxaliplatin. This approach seeks to exploit the altered metabolic state of cancer cells, providing a novel strategy to bolster traditional chemotherapy treatments.

## 2. Materials and Methods

### 2.1. Establishment of Healthy and CRC Organoid Cultures

The healthy colon organoid line, the CRC organoid lines, and the L-WRN cell line were kindly provided by the Yilmaz Lab at the Koch Center for Integrative Cancer Research, MIT, USA. The CRC organoid lines were generated by employing in vitro gene editing using CRISPR/Cas9 technology from a healthy colon organoid line, specifically introducing three primary driver mutations (Apc-/-, KRAS G12D, and TP53) implicated in carcinogenesis.

For organoid culture establishment, 1000 single cells were seeded in a 48-well flat-bottom plate using a 1:2 ratio with Matrigel (Corning, NY, USA), with a total volume of 10 μL. After a 30 min incubation at 37 °C, 300 µL organoid growth medium was added to each well and refreshed every three days. Since healthy colon cells need exogenous factors to proliferate and form organoids, the growth medium for the healthy colon organoid line was derived from the supernatant of used media from the L-WRN cell line, which is rich in Wnt-3a, RSPO1, and Noggin (WRN) growth factors. L-WRN cells were cultured in ADMEM/F12 medium supplemented with 10% FBS, devoid of antibiotics, until reaching 70% confluence, when the medium was collected. Following centrifugation of the collected WRN medium at 300 G for 5 min, the medium was stored at 4 °C. This medium was blended with 50% ADMEM/F12. The final growth medium was obtained by supplementing this mixture with 1% penicillin/streptomycin, 1X B27 (Thermo Fisher Scientific, Waltham, MA, USA), 40 ng/mL EGF (Stemcell Technologies, Vancouver, BC, Canada), and Y-27632 (Sigma Aldrich, St. Louis, MO, USA). The growth medium was refreshed every three days. Following enzymatic dissociation of the organoids, the single cells were centrifuged at 300 G for 5 min. Cancer cells do not need exogenous growth factors to generate organoids, as opposed to healthy colon cells; the growth medium for the CRC organoid line was formulated by supplementing ADMEM/F12 (Thermo Fisher Scientific, Waltham, MA, USA) basal medium with 1X B27 (Thermo Fisher Scientific, Waltham, MA, USA), 1% Penicillin-Streptomycin (Thermo Fisher Scientific, Waltham, MA, USA), and 1X Glutamax (Thermo Fisher Scientific, Waltham, MA, USA) [13,14].

### 2.2. Investigating Therapeutic Interventions and Viability in Organoid Models

After seeding the cells in Matrigel within a 48-well plate, the relevant compounds were introduced into the organoid growth medium. The IC10 and IC50 values for oxaliplatin were established by applying it to CRC organoid cultures at concentrations ranging from 0 to 21 µM (incremented by 1.5 µM), with three replicates for each concentration [15]. BOHB was evaluated at concentrations ranging from 0 to 60 µM, and melatonin was assessed in doses from 0 to 1000 µM, escalating in increments of 200 µM, on healthy colon organoid lines. These experiments were conducted in triplicate to determine the highest non-toxic doses of each compound. Furthermore, the IC10 and IC50 values of oxaliplatin, when combined with both melatonin and BOHB, were examined in double and triple combinations, respectively. All treatments were performed for 96 h, which was indicated as the maximum generation capacity for organoid lines (data are shown in Appendix A).

Cell viability was assessed by using a resazurin-based assay. Following treatment, 1X Resazurin (Cayman, Ann Arbor, MI, USA) was added to each well, and the plates were incubated at 37 °C for 4 h. Subsequently, the fluorescence intensities of each well were measured using a Varioskan Lux Plate Reader (Thermo Fisher Scientific, Waltham, MA, USA). The data from the control group were considered to represent 100% cell viability, and the viability of the other groups was evaluated relative to this benchmark. All experiments were performed in triplicate [16].

### 2.3. Protein Analysis via Western Blot

Total protein extraction from organoids was obtained using RIPA buffer (Thermo Fisher Scientific) supplemented with 1X Halt protease inhibitor (Thermo Fisher Scientific, Waltham, MA, USA). Upon performing a BCA protein assay (Thermo Fisher Scientific, Waltham, MA, USA), proteins were run on NuPage Bis-Tris gels (Thermo Fisher Scientific, Waltham, MA, USA) in an MES SDS running buffer (Thermo Fisher Scientific, Waltham, MA, USA). The proteins were then transferred onto immobilized polyvinyl difluoride membranes (Millipore Sigma, Burlington, MA, USA). After an hour-long incubation at room temperature in blocking buffer (5% BSA (Research Product International, Mt Prospect, IL, USA)), several proteins including LDHA (Abcam, Cambridge, UK), mitochondrial pyruvate carrier-2 (MPC2) (Proteintech, Rosemont, IL, USA), pyruvate dehydrogenase (PDH) (Cell Signaling Technology, Danvers, MA, USA), pyruvate dehydrogenase kinase 4 (PDK4) (Thermo Fisher Scientific, Waltham, MA, USA), cleaved caspase-3 (CC3) (Cell Signaling Technology, Danvers, MA, USA), gamma phospho-Histone H2A.X (γH2AX) (Millipore Sigma, Burlington, MA, USA), and histone H3 acetyl Lys27 (H3K27ac) (Cell Signaling Technology, Danvers, MA, USA). Antibodies were incubated at the concentrations suggested by the manufacturer at 4 °C overnight and then incubated with secondary antibodies, including rabbit anti-antibody (Abcam, Cambridge, MA, USA) and mouse anti-antibody (Abcam, Cambridge, MA, USA), at a dilution of 1:7500 for 1 h. After treating the membranes with an excellent chemiluminescent substrate kit (Advansta, San Jose, CA, USA) for 3 min, band images were captured. The bands in the images were analyzed using ImageJ software version 1.41 [17].

### 2.4. Assessment of ROS Levels

Upon treatment completion, CRC organoids were collected in microfuge tubes. Residual Matrigel was removed by centrifugation at 1000× *g* for 5 min, with the supernatant discarded. Organoids were subsequently dissociated using TrypLE for 5 min at 37 °C. ROS analysis was then performed with CellROX Orange Reagent (Thermo Fisher Scientific) according to the manufacturer’s instructions. Each sample was analyzed individually on a BD FACSAria III flow cytometer (BD Bioscience, San Jose, CA, USA), with 10,000 events. All experiments were performed in triplicate, and the data were analyzed using FlowJo software version 10.8.1 (BD Bioscience).

### 2.5. Statistical Analysis

The data obtained were evaluated and visualized using GraphPad V9. Variables gathered through methodology were summarized as mean ± standard deviation, median, and minimum–maximum, while counted variables were summarized as relative percentage distribution. The nonparametric Mann–Whitney U test was used to evaluate the significance between the two experimental groups in cell culture, while an ANOVA was employed for multiple analyses. A statistical significance level of *p* < 0.05 was deemed significant.

## 3. Results

### 3.1. Ascertaining Doses for Usage of BOHB, Melatonin, and Oxaliplatin on Organoids

A viability assay was conducted following the administration of varied doses of melatonin and BOHB to the healthy colon organoid line over a duration of 96 h. The resulting data denoted that both melatonin and BOHB, at concentrations of up to 200 µM and 10 mM, respectively, did not inflict cytotoxic effects on the healthy colon organoids, thus establishing these values as their maximum non-toxic doses, as illustrated in Figure 1A and 1B, respectively. For the determination of IC10 and IC50 values of oxaliplatin, the CRC organoid line was exposed to an array of the drug concentrations for a period of 96 h. As per the results, the IC10 and IC50 values of oxaliplatin for CRC organoids were indicated as 1.1 µM and 2.4 µM, respectively, and are illustrated in Figure 1C.

### 3.2. BOHB Enhances Oxaliplatin Treatment Efficacy in CRC Organoids

CRC organoids were treated with combination regimens of oxaliplatin (at IC10 and IC50 doses) and BOHB (ranging from 0 to 10 mM), supplemented with a constant dose of 200 µM melatonin, for a duration of 96 h. A viability assay conducted post-treatment revealed a notable enhancement in the cytotoxic efficacy of oxaliplatin on CRC organoids, as facilitated by BOHB. As the data in Figure 2A shows, at the IC10 oxaliplatin dose, this enhancement was significant with 10 mM of BOHB. Similarly, in Figure 2B, the IC50 oxaliplatin dose combined with 7.5 and 10 mM BOHB also showed augmented cytotoxicity. Under the treatment protocol consisting of IC10 oxaliplatin and 10 mM BOHB, the observed cell viability was marked at 53%. Additionally, the IC50 oxaliplatin treatments, when coupled with 7.5 mM and 10 mM BOHB, resulted in cell viability percentages of 42.99% and 37.83%, respectively. Interestingly, as shown in Figure 2C, D, the inclusion of 200 µM melatonin in the treatment regimen appeared to negate the enhancement of oxaliplatin’s cytotoxic effect mediated by BOHB.

### 3.3. BOHB Modifies Energy Metabolism Pathways

As an integral player in energy metabolism, BOHB prompted an assessment of metabolic alterations at the protein level. Fundamental proteins, namely LDHA, MPC2, PDH, and PDK4, were evaluated, offering insights into the nuanced shifts in energy metabolism pathways. Western blotting, performed 96 h post-treatment application to the CRC organoids, facilitated this examination, and the related bands are shown in Figure 3A. As illustrated in Figure 3B, control group measurements identified LDHA protein levels at 1.22, whereas the BOHB, melatonin, and their combination treatments rendered measurements of 0.75, 0.70, and 0.58, respectively. Relative LDHA protein levels fell by over four-fold in the BOHB + IC10 oxaliplatin and melatonin + BOHB + IC10 oxaliplatin groups, at 0.23 and 0.30, as compared to the control group (Figure 3C). Contrarily, MPC2 protein levels saw an uptick in the BOHB, melatonin, and combination treatment groups, at 1.07, 1.11, and 0.93, respectively, in comparison to the control group, which recorded a level of 0.49. PDH4 protein levels marginally dropped in the melatonin + BOHB + IC10 oxaliplatin group relative to other groups. Lastly, PDK4 protein levels maintained uniformity across all treatment groups. A parallel experimental configuration for IC50 oxaliplatin revealed analogous trends concerning relative LDHA protein levels (Figure 3D). Specifically, under BOHB + IC10 oxaliplatin and melatonin + BOHB + IC10 oxaliplatin treatments, LDHA protein levels displayed threefold lower expression, at 0.50 and 0.56, respectively, when compared to control groups.

### 3.4. Beta Hydroxybutyrate Does Not Influence Histone Acetylation-Dependent Apoptosis

BOHB’s role as a histone deacetylase (HDAC) inhibitor called for an evaluation of H3K27ac protein levels via Western blotting. Given that histone acetylation can foster an open chromatin form and potentially intensify the DNA adducts of oxaliplatin, the protein levels of cleaved caspase 3 (CC3) and gammaH2AX (gH2AX) were scrutinized to probe the correlation between the histone acetylation rate and the DNA damage-induced apoptosis pathway.

CRC organoids under BOHB+ IC10 and IC50 oxaliplatin treatments demonstrated a 2.1- and 2.9-fold increase in H3K27ac levels, respectively, in comparison to the control group, though the distinctions did not attain statistical significance (Figure 4). Likewise, CC3 and gH2AX protein levels exhibited no considerable disparities when BOHB+ IC10 and IC50 oxaliplatin were juxtaposed with IC10 and IC50 oxaliplatin in isolation. Notably, the highest gH2AX protein levels were associated with the BOHB+ oxaliplatin treatment.

### 3.5. Melatonin Shields Colorectal Cancer Cells from ROS-Induced Apoptosis

In triple combinations, melatonin appears to mitigate the enhanced cytotoxicity of oxaliplatin propelled by BOHB. To delve into the protective mechanism of melatonin on cancer cells and its plausible modulatory influence on the apoptotic pathway, the study scrutinized levels of the apoptotic marker CC3, the DNA damage marker gH2AX, and ROS across diverse treatment scenarios.

The viability analysis across all combination and single treatment applications hints at melatonin functioning as an antagonist to BOHB-enhanced oxaliplatin cytotoxicity. Numerous investigations identify melatonin as a ROS scavenger agent, potentially elucidating the observed decrease in oxaliplatin cytotoxicity following melatonin treatment. Alternatively, melatonin may bolster the survival capacity of cancer cells by inhibiting the DNA damage and apoptosis pathway. As exhibited in Figure 5, comparable relative expression levels were observed in the melatonin treatment and control groups for the apoptotic marker CC3 (0.78033 and 0.75413, respectively), as well as for the DNA damage marker gH2AX (0.3346 and 0.31436, respectively). Oxaliplatin, a known inducer of apoptosis via DNA damage, was linked to elevated levels of CC3 and gH2AX expression in both IC10 and IC50 treatment groups. However, the amalgamation of melatonin and oxaliplatin treatment noticeably diminished the expression levels of both markers, suggesting a reduction in DNA damage and apoptotic cell death. These results provide a mechanistic elucidation for the observed protective impact of melatonin against oxaliplatin toxicity.

ROS levels were gauged in each treatment group using flow cytometry. As indicated in Figure 5C, a surge in ROS positivity was noted in live cells subjected to melatonin treatment, compared to other treatment groups. The mel + IC50 oxa, mel + BOHB, and mel + BOHB + IC50 oxa treatment groups displayed the highest proportion of ROS-positive live cells, with values of 21.7667%, 21.5333%, and 19.8%, respectively. In contrast, the BOHB + IC10 oxa treatment group exhibited the lowest proportion of ROS-positive live cells, at 6.8%.

## 4. Discussion

The therapeutic potential of addressing tumor cells’ altered metabolism is a significant and promising avenue in cancer treatment [18,19]. This study aimed to delineate the highest non-toxic doses of neo-adjuvants on healthy colon organoids, thereby reducing any adverse toxic effects related to cancer treatment. For this purpose, the maximum non-toxic dose for BOHB was found to be 10 mM, and it was 200 µM for melatonin.

While BOHB is typically found at levels around 0.5 mM in circulation, it is known to increase to 2 mM [20,21] following two days of fasting, and up to 8 mM during prolonged starvation [22]. It is noteworthy that even at 10 mM, BOHB demonstrated no cytotoxicity on healthy colon organoids. This aligns with a recent study by Dmitrieva-Posocco O. et al., further validating the safety of BOHB at these concentrations. They applied BOHB concentrations ranging from 0 to 25 mM and observed no cytotoxic effect on healthy colon organoids up to 10 mM. However, doses of 15 mM and higher induced cytotoxicity on healthy colon organoids. This finding aligns with the results of the current research [23].

Melatonin was used in this study to elucidate the BOHB-mediated oxidative stress that leads to apoptosis. In a study by Guangyu Ji. et al., melatonin exhibited cytotoxicity at 1 mM in a wide range of colon cancer lines. Prior studies have been conducted in two-dimensional culture systems and on cancer cell lines; hence, the non-toxic dose of melatonin in three-dimensional and healthy colon organoid cultures was determined for the first time in this research [24,25,26].

In combination therapy applications, 10 mM BOHB + IC10 oxa, and 7.5 mM and 10 mM BOHB + IC50 oxa demonstrated a statistically significantly enhanced cytotoxicity on CRC organoid culture, compared to treatments with oxaliplatin only. While dual treatments with melatonin and oxaliplatin showed no significant difference in treatment efficacy, melatonin neutralized the enhanced cytotoxicity of oxaliplatin mediated by BOHB. Collectively, these findings shed light on the potential of utilizing BOHB in combination therapies with oxaliplatin, which warrants further research. This study lays the groundwork for exploring these treatment modalities’ impacts on CRC, marking an important step in understanding their therapeutic potential. This is the first study to assess the effect of BOHB in combination with oxaliplatin treatment on CRC. Contrastingly, when healthy colon organoids were treated with BOHB (at concentrations of 2.5, 5, 7.5, and 10 mM) combined with IC10 oxa, viability assessments revealed no significant decrease across any treatment conditions (the data are shown in Appendix A). This indicates that the combination therapy selectively enhances cytotoxicity in CRC organoids alone.

Multiple factors could contribute to the observed enhancement of oxaliplatin cytotoxicity. One possibility is that BOHB suppresses the glycolytic pathway, thereby inhibiting the Warburg effect. Additionally, as BOHB is used as a source of acetyl-CoA, it could potentially cause an upsurge in the TCA cycle, thus increasing the volume of ROS produced by ETC components. This, in turn, could have contributed to elevated ROS levels, thereby augmenting oxaliplatin toxicity. Lastly, given BOHB’s known role as a HDAC inhibitor, it may push cancer cells towards apoptosis by promoting the DNA binding potential of oxaliplatin through the formation of an open chromatin structure.

One key element in aerobic glycolysis or the Warburg effect is lactate dehydrogenase A (LDHA), which regenerates the electron acceptor NAD+ and is widely considered an appealing target for cancer therapeutics [27]. This study demonstrated that BOHB decreased LDHA and MPC2 protein levels compared to the control group, suggesting that BOHB may channel energy metabolism through the TCA cycle.

Although a link was presented between the function of the TCA cycle and MPC protein in prostate cancer by Bader D. A. et al. [28], no previous research has examined the relationship between BOHB and the LDHA-MPC2 proteins. Despite indications that BOHB might influence energy pathways, prior research found no evidence that it impacts the therapeutic efficacy in breast cancer cell lines [29]. Another study showed the results that inhibiting LDHA correlated with oxidative stress-mediated suppression of tumor progression in certain lymphoma cell lines [30].

Notably, the ROS measurements in this study revealed that the groups treated with BOHB had the lowest ROS levels at the end of the treatment. When considered alongside the cell viability results, it seems that the increased toxicity could be ROS-dependent. A recently published study by Chen W. et al. showed that oxaliplatin treatment efficacy is elevated by enhancing oxidative stress with piperlongumine in vitro and in vivo. This study supports the results of our research, that mechanistically oxidative stress contributes to oxaliplatin cytotoxicity in CRC cells [31].

Numerous in vitro and in vivo studies have demonstrated that melatonin can reduce oxidative stress-dependent damage to proteins, lipids, and DNA [32]. Leveraging this antioxidative property of melatonin in the present study, it was found that treatment groups with melatonin showed the highest percentages of ROS-positive viable cells. This finding suggests that melatonin may serve a protective role, shielding cells from death mechanisms under high oxidative stress. Further supporting this notion, levels of the DNA damage marker gH2AX and the apoptotic marker CC3 were lower in cells treated with melatonin and oxaliplatin, compared to those treated with oxaliplatin alone. This is indicative of melatonin’s potential to inhibit DNA damage-mediated apoptosis and protect against oxaliplatin toxicity.

In line with this, a study by Waseem M. et al. found that melatonin exerts a protective effect against oxaliplatin-induced mitochondrial stress and apoptotic cell death in a human SYSY-5Y neuroblastoma cell line [33]. In another investigation, it was revealed that melatonin in combination with 5-fluorouracil suppresses stem cells in colon cancer via cellular prion protein-Oct4 axis regulation [34]. Several studies have documented the cytotoxic effect of melatonin on CRC cell lines, but it should be noted that these studies used doses ranging from 1 to 5 mM [15], which are considerably high and may cause off-target effects on cell viability.

Furthermore, BOHB is recognized as an HDAC inhibitor. This attribute could potentially enhance the DNA adduct formation of oxaliplatin, given that acetylated DNA generally exhibits a more open chromatin form [35]. Despite observing a threefold increase in H3K27ac protein levels, the lack of statistical significance is likely attributable to the sample size (Figure 4). Interestingly, existing research illustrates how BOHB can enhance cisplatin cytotoxicity by inhibiting the HDAC/survivin axis in the HEPG2 cell line, a human hepatocellular carcinoma cell line [36].

As a result, the alteration in energy metabolism, induced by BOHB administration, led to an increase in ROS levels, thereby enhancing the efficacy of oxaliplatin treatment on CRC organoids. Melatonin counteracted this enhanced oxaliplatin cytotoxicity by inhibiting DNA damage and promoting apoptotic pathways under conditions of high oxidative stress induced by BOHB treatment.

This study comes with several limitations, including a lack of analysis on the effects of BOHB and melatonin on DNA repair, and other cell death mechanisms such as autophagy, necrosis, and ferroptosis pathways. Within the limitations of the model, crucial microenvironmental elements impacting treatment effects were absent. Another limitation of the study is not involving in vivo drug metabolism, as seen in more realistic models, which was not considered in this study.

## 5. Conclusions

In conclusion, this study elucidates the role of BOHB-mediated energy metabolism alteration in enhancing the efficacy of oxaliplatin treatment in colorectal cancer (CRC) organoids, offering invaluable insights into the potential merits of this therapeutic approach.

These findings not only pave the way for an effective means to augment existing treatment regimens but also shed light on the intricate interplay between energy metabolism and drug response in CRC. By pinpointing the synergistic interaction between BOHB and oxaliplatin, the study emphasizes the necessity of exploring innovative drug combinations with the aim of optimizing therapeutic outcomes while minimizing potential adverse effects. These results could prompt the development of more personalized and targeted treatment strategies for CRC patients, ultimately enhancing their prognosis and quality of life.

While additional research is required to thoroughly delineate the mechanisms underpinning BOHB’s impact on energy metabolism and its potential implementation in clinical settings, the present study unquestionably sets the groundwork for future advancements in cancer therapy. The outcomes of this research provide a persuasive argument for investigating the integration of BOHB into existing therapeutic strategies for CRC, underscoring the importance of pursuing innovative drug combinations in the continuous battle against cancer. As we persist in deciphering the complexities of cancer biology and metabolism, these revelations hold significant promise for shaping the future landscape of oncology and enhancing patient outcomes.

## Figures and Tables

**Figure 1 cancers-15-05724-f001:**
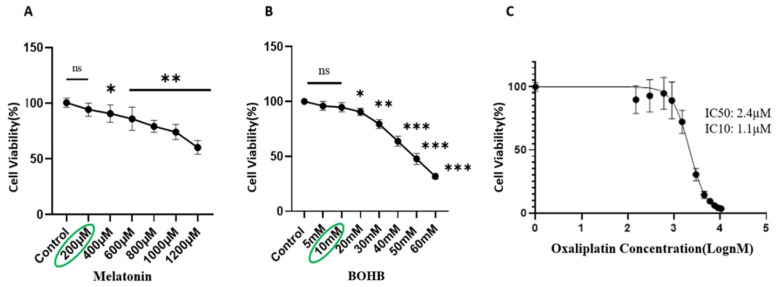
Assessment of cytotoxic effects of melatonin, BOHB, and oxaliplatin on normal colon organoids. Single cells derived from normal colon organoids were cultured in 48-well plates and subjected to increasing concentrations of melatonin (ranging from 0 to 1200 µM), BOHB (ranging from 0 to 60 mM), and oxaliplatin (ranging from 0 to 21 µM, in increments of 1.5 µM) as depicted in panels (**A**–**C**), respectively. Following 96 h of treatment, cell viability was evaluated using a resazurin assay. All experiments were performed in fourfold replication, both technically and biologically. Cytotoxic impacts of the treatments are demonstrated as percentages relative to the untreated controls (100%). Asterisks indicate treatments with significant cytotoxic effects (*p* > 0.05 no significant (ns), * *p* < 0.05, ** *p* < 0.01, *** *p* < 0.001 compared to the control).

**Figure 2 cancers-15-05724-f002:**
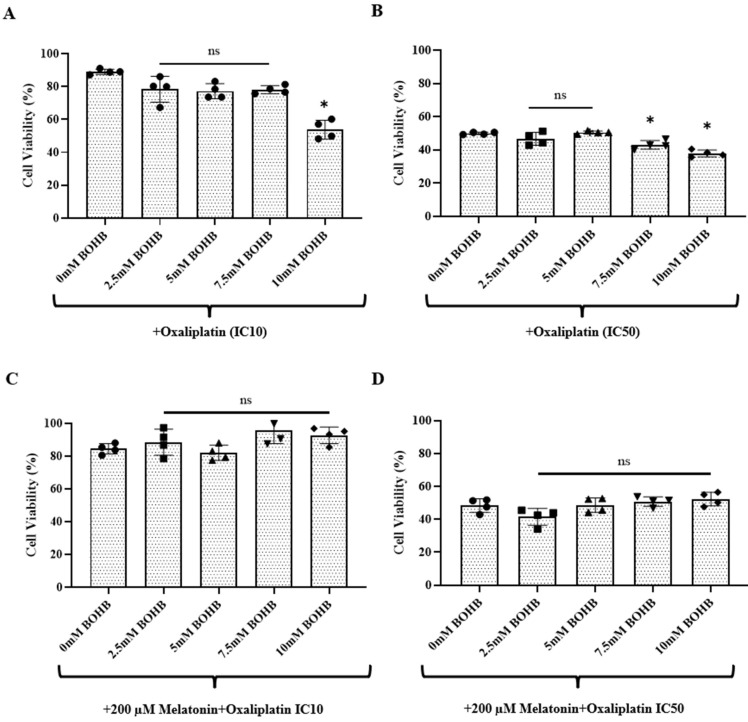
Potentiation of oxaliplatin treatment efficacy by beta-hydroxybutyrate in colorectal cancer organoids. Single cells dissociated from colorectal cancer organoids were seeded in 48-well plates and exposed to increasing concentrations of BOHB (0–10 mM) at IC10 and IC50 oxaliplatin doses, as shown in panels (**A**,**B**). Treatments also included 200 µM of melatonin in combination with these dual therapies, as depicted in panels (**C**,**D**). Cell viability was quantified using a resazurin assay after 96 h of treatment. All experiments were performed in quadruplicate, both technically and biologically. The cytotoxic impact of the treatments is illustrated as a percentage of the cell viability observed in untreated controls (represented as 100%). The asterisk indicates treatments exhibiting statistically significant variations (*p* > 0.05 no significant (ns), *p* < 0.05 significant, relative to the control).

**Figure 3 cancers-15-05724-f003:**
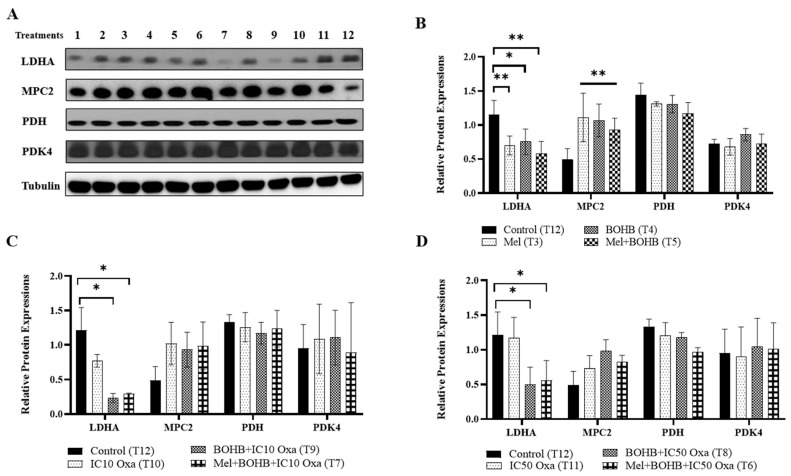
Induction of metabolic pathway alterations by beta-hydroxybutyrate. This figure outlines the changes in the expression levels of proteins related to energy metabolism under various treatment conditions. (**A**) A series of treatments, labeled 1 to 12, were administered to colorectal cancer organoids, incorporating components such as 10 mM BOHB, 200 µM melatonin, and IC10-IC50 oxaliplatin: 1. melatonin + IC10 oxaliplatin, 2. melatonin + IC50 oxaliplatin, 3. melatonin, 4. BOHB, 5. melatonin + BOHB, 6. melatonin + BOHB + IC50 oxaliplatin, 7. melatonin + BOHB + IC10 oxaliplatin, 8. BOHB + IC50 oxaliplatin, 9. BOHB + IC10 oxaliplatin, 10. IC10 oxaliplatin, 11. IC50 oxaliplatin, 12. control. Post 96 h treatment application to colorectal cancer organoids within a 48-well plate setup, cell lysates were subsequently harvested. The uncropped blots are shown in the Appendix A. (**B**) The bar graph elucidates the expression levels of LDHA, MPC2, PDH, and PDK4 proteins under treatments 3, 4, and 5. (**C**) The bar graph exhibits the expression profiles of LDHA, MPC2, PDH, and PDK4 proteins under treatments 7, 9, and 10 (pertaining to IC10 oxaliplatin-inclusive treatments). (**D**) The bar graph portrays the expression levels of LDHA, MPC2, PDH, and PDK4 proteins under treatments 6, 8, and 11 (associated with IC50 oxaliplatin-inclusive treatments). Protein levels were quantified using ImageJ software version 1.41, with normalization performed against tubulin levels. All experiments were carried out with three biological and five technical replicates. Asterisks symbolize the statistical significance of the treatments (* *p* < 0.05, ** *p* < 0.01 relative to the control).

**Figure 4 cancers-15-05724-f004:**
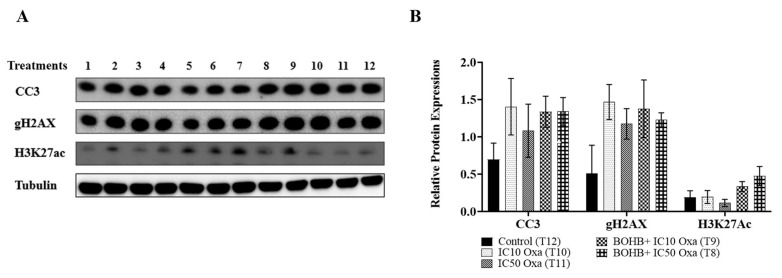
Examination of changes in protein expression levels related to cell death mechanism and histone acetylation under various treatments. (**A**) Colorectal cancer organoids underwent distinct treatments, marked from 1 to 12, which included 10 mM BOHB, 200 µM melatonin, and IC10-IC50 oxaliplatin: 1. melatonin + IC10 oxaliplatin, 2. melatonin + IC50 oxaliplatin, 3. melatonin, 4. BOHB, 5. melatonin + BOHB, 6. melatonin + BOHB + IC50 oxaliplatin, 7. melatonin + BOHB + IC10 oxaliplatin, 8. BOHB + IC50 oxaliplatin, 9. BOHB + IC10 oxaliplatin, 10. IC10 oxaliplatin, 11. IC50 oxaliplatin, 12. control. The treatments were applied for 96 h to the organoids in a 48-well plate, after which cell lysates were collected. Western blotting was then used to analyze the protein levels of CC3, gH2AX, and H3K27ac. The uncropped blots are shown in Appendix A (**B**) The bar graphs depict the protein levels of CC3, gH2AX, and H3K27ac under treatments T8, T9, T10, and T11. Protein levels were quantified in ImageJ software version 1.41, with normalization performed by dividing by tubulin levels. All experiments were performed with three biological and five technical replicates.

**Figure 5 cancers-15-05724-f005:**
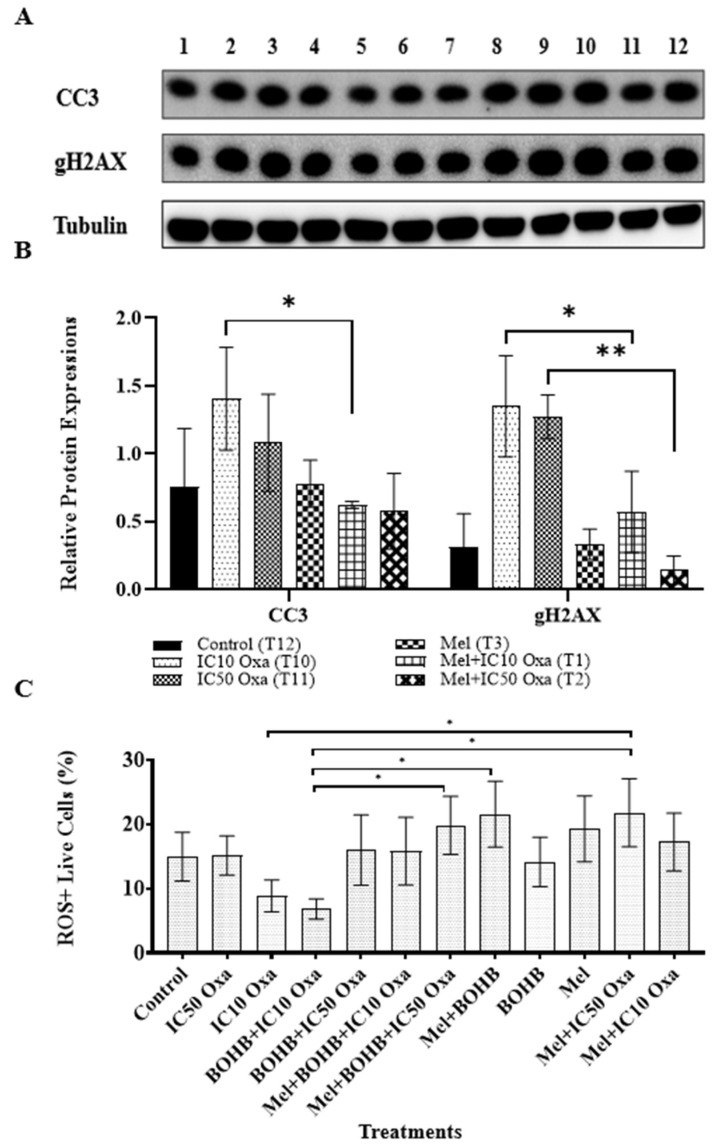
Analysis of melatonin’s ability to inhibit apoptotic cell death induced by oxaliplatin-mediated DNA damage. (**A**) Colorectal cancer organoids underwent various treatments, labelled 1 to 12, which included 10 mM BOHB, 200 µM melatonin, and IC10-IC50 oxa: 1. melatonin + IC10 oxa, 2. melatonin + IC50 oxa, 3. melatonin, 4. BOHB, 5. melatonin + BOHB, 6. melatonin + BOHB + IC50 oxa, 7. melatonin + BOHB + IC10 oxa, 8. BOHB + IC50 oxa, 9. BOHB + IC10 oxa, 10. IC10 oxa, 11. IC50 oxa, 12. control. The treatments were administered to the organoids in a 48-well plate over 96 h, after which the cell lysates were gathered. The protein levels of CC3 and gH2AX were subsequently assessed via Western blotting. The uncropped blots are shown in Appendix A. (**B**) The bar graphs depict the protein levels of CC3 and gH2AX under treatments T1, T2, T3, T10, and T11. Protein levels were quantified in ImageJ software version 1.41, with tubulin levels serving as the normalization control. (**C**) After a 96 h treatment period, the collected colorectal cancer organoids were dissociated into single-cell forms, and the cells were subjected to an ROS detection kit. The graphic represents the percentage of ROS-positive live cells. Each treatment was analyzed individually by a flow cytometer, with ten thousand events recorded per treatment. All experiments were carried out with three biological and five technical replicates. Asterisks indicate the statistical significance of the treatments (* *p* < 0.05, ** *p* < 0.01, compared to the control).

## Data Availability

The data presented in this study are available in this article.

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
