# Peer review of "Beta-Hydroxybutyrate Augments Oxaliplatin-Induced Cytotoxicity by Altering Energy Metabolism in Colorectal Cancer Organoids"

_cancers, 2023, doi:10.3390/cancers15245724_

Round 1

Reviewer 1 Report

Comments and Suggestions for Authors

In this manuscript, the authors investigated the impact of beta-hydroxybutyrate (BOHB) on the cytotoxic effects of oxaliplatin in colorectal cancer (CRC) organoids. Treatment with BOHB was found to enhance the efficacy of oxaliplatin in CRC organoids. Additionally, the combined treatment of BOHB and oxaliplatin led to an increase in reactive oxygen species (ROS) production when compared to oxaliplatin treatment alone. The authors posited that BOHB treatment altered energy metabolism and increased ROS levels, thereby enhancing oxaliplatin's efficacy in CRC organoids. However, there are several points that require clarification:

1. The authors suggested that alterations in energy metabolism and ROS are potential mechanisms through which BOHB increases oxaliplatin's cytotoxic effect. However, it appears that the enzymes involved in energy metabolism and the proportion of ROS-positive cells were not affected by the addition of melatonin. These results suggest that the mechanism by which melatonin suppresses the effect of BOHB on oxaliplatin's cytotoxicity may have little to do with energy metabolism and ROS. If the authors contend that ROS mediates the effects of BOHB, we recommend conducting experiments using antioxidants or equivalent studies to provide further support.

2. In the Graphical Abstract, melatonin is depicted as suppressing ROS. However, in Figure 5, ROS did not appear to influence the proportion of ROS-positive cells. Therefore, we recommend correcting the Graphical Abstract to align with the observed results.

3. It would be helpful for readers if the authors added figure numbers to the Results section for easy reference.

Author Response

Dear Reviewers,

First and foremost, we would like to express our sincere gratitude for the time, effort, and expertise you have dedicated to reviewing our manuscript. Your invaluable criticisms and feedback have played a pivotal role in enhancing the quality of our study.

Each comment provided by you was thoroughly considered and addressed in the order it was received. We believe that your input has shed light on areas that might have been ambiguous or unclear. It is through this rigorous review process and your discerning insights that we have been able to refine our manuscript to its current form.

We understand the importance of maintaining the integrity and quality of research, and we deeply appreciate your commitment to this goal. Your feedback not only helps us but also contributes to the advancement of our field.

Thank you once again for your contribution. We truly value the expertise and perspective you have provided.

Regards

Gizem Calibasi-Kocal

Reviewer 2 Report

Comments and Suggestions for Authors

This study is the first to assess the effect of beta hydroxybutyrate (BOHB) in combination with oxaliplatin treatment on colorectal cancer (CRC) organoids and suggests that BOHB, by altering metabolic state, could increase the effectiveness of chemotherapy in CRC patients.

This paper raised some questions and concerns. Some results are not convincing enough.

- Please provide some information about the heathy colon organoid line used (and/or some references).  Only one colon organoid cell line used ? Coming from normal colon at distance from the tumor of a CRC patient ? What are the features of the organoids used : epithelial cell types present in the organoids ?

- “CRC organoid lines” : it seems that they are derived from the normal colon organoid line edited by Crispr/Cas9 for Apc, Kras and TP53. Why the authors did not generate organoids directly from biopsies or surgical resections of CRC patients with known mutation status (and sensitivity to chemotherapy) ?

-The L-WRN cell line is used for providing conditioned medium rich on Wnt3a, R-spondin and Noggin, used for culturing normal colon organoids and not CRC organoids ?

- It seems that some experiments were performed in 2D with cells derived from organoids (normal colon) and others on 3D organoids (CRC). Why the authors did not use 3D organoids in all experiments ?

What is the penetration depth of compounds into the 3D organoids?

- It is somewhat difficult to go through the figures showing western blots and quantification of the protein levels.

- ROS measurements by flow cytometry (Figure 5) : please show your gating strategy and some representative dotplots.  

- It is necessary to appropriately cite figures throughout the text.

Author Response

(The authors gave the same response as above.)

Reviewer 3 Report

Comments and Suggestions for Authors

Authors have explored whether the cytotoxicity of oxaliplatin towards CRC organoids might be increased with non-toxic concentrations of BOHB. I found their hypothesis of great interest and consider that have obtained promising results. 

However, there are certain issues that, in my opinion, might enhance the overall quality of the present work. First of all, there exists plenty of works to enhance the effectivity of oxaliplatin. One of the main issues of chemotherapy is not its effectivity but its side effects. Therefore, novel approaches must maintain effectivity while reducing side effects. Authors show that the combination of non-toxic concentrations of BOHB and IC10 of cisplatin results in aprox a reduction in half of CRC organoids viability, whereas the combination of IC50 and 10 mM BOHB results in slight differences in comparison with IC50 alone. Which are the effect of both combinations on healthy colon organoids? Toxicity on non-cancer tissues must be included, since I consider this parameter more relevant than toxicity on cancer tissues at this moment. In line with this, on the Introduction section more data regarding the main side effects of oxaliplatin should be included, to better explain why is it interesting to reduce its therapeutic dose. 

Furthermore, an isobologram should be included to determine whether the combined effect of oxaliplatin and BOHB responds to addition or synergy. 

Finally, Figure 1 is duplicated and should be removed the one before Introduction section.

Author Response

(The authors gave the same response as above.)

Round 2

Reviewer 1 Report

Comments and Suggestions for Authors

The authors have satisfactorily addressed the points which I noted.

Reviewer 2 Report

Comments and Suggestions for Authors

The authors took into account most of the issues I raised.

Reviewer 3 Report

Comments and Suggestions for Authors

Authors have indeed taken in consideration my review report and have included my suggestions and/or argumented why shouldn't be included. I suggest that the manuscript should be accepted for publication in its present form.